# Potential of Hyperspectral and Thermal Proximal Sensing for Estimating Growth Performance and Yield of Soybean Exposed to Different Drip Irrigation Regimes Under Arid Conditions

**DOI:** 10.3390/s20226569

**Published:** 2020-11-17

**Authors:** Adel H. Elmetwalli, Salah El-Hendawy, Nasser Al-Suhaibani, Majed Alotaibi, Muhammad Usman Tahir, Muhammad Mubushar, Wael M. Hassan, Salah Elsayed

**Affiliations:** 1Agricultural Engineering Department, Faculty of Agriculture, Tanta University, Tanta 31527, Egypt; adel.elmetwali@agr.tanta.edu.eg; 2Department of Plant Production, College of Food and Agriculture Sciences, King Saud University, Riyadh 11451, Saudi Arabia; nsuhaib@ksu.edu.sa (N.A.-S.); malotaibia@ksu.edu.sa (M.A.); mtahir@ksu.edu.sa (M.U.T.); mmubushar@ksu.edu.sa (M.M.); 3Department of Agronomy, Faculty of Agriculture, Suez Canal University, Ismailia 41522, Egypt; 4Department of Biology, College of Science and Humanities at Quwayiyah, Shaqra University, Riyadh 19257, Saudi Arabia; wmohamed@su.edu.sa; 5Department of Agricultural Botany, Faculty of Agriculture, Suez Canal University, Ismailia 41522, Egypt; 6Agricultural Engineering, Evaluation of Natural Resources Department, Environmental Studies and Research Institute, University of Sadat City, Minufiya 32897, Egypt; salah.emam@esri.usc.edu.eg

**Keywords:** canopy water mass, crop phenotyping, 2-D correlograms CWSI, growth stages, NRCT, visible/near-infrared spectroscopy, water stress

## Abstract

Proximal hyperspectral sensing tools could complement and perhaps replace destructive traditional methods for accurate estimation and monitoring of various morpho-physiological plant indicators. In this study, we assessed the potential of thermal imaging (TI) criteria and spectral reflectance indices (SRIs) to monitor different vegetative growth traits (biomass fresh weight, biomass dry weight, and canopy water mass) and seed yield (SY) of soybean exposed to 100%, 75%, and 50% of estimated crop evapotranspiration (ETc). These different plant traits were evaluated and related to TI criteria and SRIs at the beginning bloom (R1) and full seed (R6) growth stages. Results showed that all plant traits, TI criteria, and SRIs presented significant variations (*p* < 0.05) among irrigation regimes at both growth stages. The performance of TI criteria and SRIs for assessment of vegetative growth traits and SY fluctuated when relationships were analyzed for each irrigation regime or growth stage separately or when the data of both conditions were combined together. TI criteria and SRIs exhibited a moderate to strong relationship with vegetative growth traits when data from different irrigation regimes were pooled together at each growth stage or vice versa. The R6 and R1 growth stages are suitable for assessing SY under full (100% ETc) and severe (50% ETc) irrigation regimes, respectively, using SRIs. The overall results indicate that the usefulness of the TI and SRIs for assessment of growth, yield, and water status of soybean under arid conditions is limited to the growth stage, the irrigation level, and the combination between them.

## 1. Introduction

Irrigated agricultural lands play an important role in stabilizing global food security. Although these lands cover only 20% of the world’s cultivated land, they are responsible for approximately 40% of global food production. Irrigated lands in developing countries account for one-fifth of the total arable area but produce three-fifths of cereal production and constitute two-fifths of all crops [1]. At least 80% of cultivated lands in arid and semiarid countries rely on irrigation. However, moisture stress is one of the widespread abiotic factors limiting plant growth and productivity of major crops under irrigated agriculture. Additionally, abrupt climate change, high temperature and evaporation rates, and sparse and highly variable rainfall further exacerbate the drought-stress problems in irrigated agriculture, which poses a serious threat to future global food security [2,3,4]. Consequently, maximizing production per unit of irrigation water applied by shifting the irrigation strategies of crops from the paradigm of full irrigation to deficit irrigation practices is one plausible solution for addressing water shortage in irrigated agriculture [5,6].

Soybean (*Glycine max* (L.) Merr.) is the most important oilseed legume crop worldwide, and it is the fourth largest in terms of cultivated area in the world. It is considered a good source of edible oil for humans and protein concentrates for food uses, animal feed, and various industrial products. It is expected that there will be a need to expand the cultivated area of this crop in the coming years to face the demand for edible oil and protein. Therefore, this crop is already playing an important role in the food chain and will continue to do so in the future. However, moisture stress caused by insufficient water supply is one of the major problems adversely affecting the growth and productivity of soybean. The seed yield (SY) of soybean is most susceptible to moisture stress during the productive period, especially during the flowering-pod setting and seed-filling periods. However, long-term, severe moisture stress during the vegetative period may also be great enough to cause a substantial reduction in SY [7,8,9]. Several field studies have reported that exposure of soybean to moisture stress during the productive period can lead to over 50% reduction in SY [10,11,12,13].

Generally, moisture stress directly affects plant water status, which can interfere with normal plant functions and can cause substantial changes in morphological and physiological traits. These effects subsequently translate into substantial changes in crop growth and development with significantly decreased leaf area, biomass accumulation, photosynthesis efficiency, stomatal conductance, transpiration rate, and chlorophyll content and ultimately significant losses in SY [4,7,14,15,16]. These reports indicate that the close association between plant water status and available soil water can be exploited to effectively manage irrigation scheduling that involves deciding how much and when irrigation water should be applied and how to simultaneously maximize yield and water use efficiency through accurate estimation and monitoring of the various morpho-physiological indicators [17,18,19,20,21]. Although the monitoring and assessment of common morpho-physiological indicators that reflect crop water stress or vegetation water status using traditional plant sampling techniques are very accurate and simple and do not require special expertise, they are generally tedious, destructive, time- and cost-inefficient, and often inappropriate for multi-scale and real-time monitoring of these indicators on a large scale.

In recent years, quick, nondestructive, and low-cost phenotyping tools such as thermal imaging cameras and proximal passive reflectance sensors have been demonstrated to be a powerful alternative to traditional plant sampling techniques for the integrative assessment of multiple morpho-physiological indicators that accurately reflect crop water stress. Interestingly, these tools are feasible when crop water-stress indicators are evaluated under several field treatments and on a large scale.

The earliest response to moderate and severe soil moisture stress is stomatal closure to avoid excessive water loss through transpiration, which occurs before any change in the root system, leaf area, leaf water potential, and/or relative water content [22,23,24,25]. For example, stomatal conductance decreased by 60% under drought stress in soybean [26,27]. In another study, the decrease in stomatal conductance reached 92% under severe drought stress in soybean [28]. A decrease in stomatal conductance for an extended period leads to a considerable increase in canopy temperature due to the limited ability of the transpiration process to cool the canopy. Therefore, the infrared thermal imaging tool can effectively monitor water status based on the whole-canopy temperature. This tool measures the canopy reflectance of infrared radiation and uses the measurement to calculate the spatial distribution of the canopy temperature [25,29,30]. The monitoring of canopy temperature by thermal imaging is nondestructive and scalable from a single plant to whole-field traits and can be performed several times over the growing season [31,32]. Additionally, a time series of canopy temperature can effectively identify the growth stages sensitive to a lack of water in the root zone.

The crop water-stress index (CWSI) is the most commonly used crop index that estimates crop water stress based on the canopy surface temperature and thermal data [33]. Several previous studies have reported a strong correlation between CWSI, and crop water stress and yield in different field crops [15,34,35]. The normalized relative canopy content (NRCT), developed by Elsayed et al. [19,30] and based on the actual infrared temperature of the canopy, has been recognized as an indicator of crop water stress in some barley varieties.

Proximal remote sensing has the potential to complement and perhaps replace traditional methods in estimating and monitoring several phenotypic plant traits and can facilitate robust, quick, cost effective, and nondestructive estimation. It can track even slight changes induced by moisture stress that take place in different biophysical and biochemical characteristics of the canopy in the visible (VIS) 400–700 nm to near-infrared (NIR) 700–1300 nm and shortwave-infrared (SWIR) 1300–2500 nm ranges of the electromagnetic spectrum. Generally, the changes that take place in the characteristics of leaf mesophyll, biomass accumulation, leaf pigments and nutrient concentrations, leaf area index (LAI), photosynthetic efficiency, and leaf water content result in unique changes in the spectral signatures of the electromagnetic spectrum reflected from the canopy [36,37,38,39,40,41]. For example, leaf pigments such as carotenoids, chlorophylls, xanthophylls, and anthocyanins strongly absorb radiation in the VIS region, especially in the blue and red wavebands, but not in the NIR region of the spectrum. The magnitude of canopy spectral reflectance in the NIR region of the spectrum is strongly influenced by the scattering and diffusion of radiation due to leaf dry matter content and different leaf tissues. The absorption features of leaf water content are primarily located in the NIR and SWIR regions of the spectrum, with the peaks of water absorption bands located in the SWIR regions near 1450, 1940, 2100, 2250, and 2500 nm, while the weak absorption bands are identified in the NIR region near 970 and 1200 nm. The absorption peak in the SWIR region can be attributed to the combinations and overtones of the fundamental vibrational excitation within the water molecule (O–H) [4,42,43]. However, weak absorption bands, especially at 970 nm, can be attributed to the ability of NIR wavebands to penetrate more deeply into the canopy than SWIR wavebands and, therefore, provide the ability to estimate water content in the entire canopy more accurately than SWIR [17,44]. Due to the close relationship between specific wavelengths in these three parts of the electromagnetic spectrum (VIS, NIR, and SWIR) and the different biophysical and biochemical characteristics of the canopy, these specific wavelengths have been exploited to calculate specific spectral reflectance indices (SRIs) using simple mathematical operations (normalized or ratio formulas). These SIRs have been employed in estimating and monitoring several plant phenotypic traits such as aboveground dry and fresh biomass, grain yield and its components, pigment contents, and most plant measurements that reflect the status of plant water [6,37,43,45,46,47,48].

Little attention has been given to assessing the combination of thermal and spectroradiometric measurements in detecting plant growth, crop water status, and yield of soybean under arid climatic conditions. Therefore, the present study aimed (1) to quantify the response of different vegetative growth traits and SY of soybean to different irrigation regimes at two growth stages and (2) to examine the potential of thermal infrared measurements via CWSI and NRCT and of spectroradiometric measurements using published and newly constructed SRIs to indirectly estimate different soybean traits under different irrigation regimes and growth stages specifically or at each growth stage across irrigation regimes or vice versa.

## 2. Materials and Methods

### 2.1. Experimental Site and Conditions

The field experiment on drip-irrigated soybean was conducted at the Experimental Research Station (30°2′41.2″ N, 31°14′8.2″ E) during the summer seasons of 2016 and 2017. The experimental site conditions were an arid climate with a dry summer season and high temperature. The minimum and maximum temperatures during the summer growing season (June to October) ranged from 28.3 °C to 40.2 °C. The soil texture of the experimental site is sandy loam, consisting of 72.8% sand, 19.3% silt, and 7.9% clay; bulk density 1.45 g cm^−3^; electrical conductivity 1.12 dS m^−1;^ field capacity 19.2%; wilting point 10.1%; and available water 9.1%. The quality of water used for irrigation was normal, with an electrical conductivity of 1.20 dS m^−1^.

### 2.2. Experimental Design and Agronomic Practices

A randomized complete block design with four replications and three irrigation regime treatments was used for each growing season. The replications and irrigation treatments resulted in 12 plots. The irrigation treatments were randomly assigned to each plot within a replicate. The layout of the experimental plots is illustrated in Figure 1. The plot areas were 96.0 m^2^ each, and each plot was supplied with four polyethylene lateral drip lines 16 mm in diameter and 32 m long, with a 0.30 m emitter spacing (Chapin Watermatics, Watertown, NY, USA). The lateral drip line was laid out along each soybean row 0.75 m apart. The drippers had a discharge rate of 4 l h^−1^. Each lateral line was equipped with a T-shape valve to control water flow from the sub-mainlines (50 mm in diameter) to the lateral drip lines.

Four seeds of the cultivar Giza 111, belonging to the indeterminate growth type and group maturity IV of soybean varieties, were sown around each dripper on April 24 in both growing seasons. At two weeks after sowing, the seedlings were thinned to two plants per dripper to obtain a final plant population of approximately 88,889 plants ha^−1^. The entire dose of phosphorus (50 kg P_2_O_5_ ha^−1^) was applied basally as superphosphate (15.5% P_2_O_5_) before sowing. All treatments received 100 kg N ha^−1^ as ammonium nitrate (33.5% N) applied by fertigation in five equal doses starting from sowing to the full bloom stage. Additionally, 60 kg K_2_O ha^−1^ was applied 35 days after sowing in two equal doses, with 15-day intervals between doses. All other recommended agronomic practices (control of weeds, diseases, and pests) were performed in a timely manner. Hand harvesting was performed on 10 August 2016 (108 days after sowing) and 14 August 2017 (112 days after sowing).

All irrigation treatments received the same volume of irrigation water at 4-day intervals (approximately 60 mm, starting from sowing to three weeks after sowing) to ensure the complete establishment of seedlings. Thereafter, irrigation water was applied according to the prescribed irrigation treatments outlined below.

### 2.3. Irrigation Treatments

In this study, three irrigation treatments were applied, 100%, 75%, and 50% of the estimated crop evapotranspiration (ETc), with a 4-day irrigation interval. The irrigation water volume (ETc) for the full irrigation treatment (100% ETc) was estimated using the following equation:(1)ETc=ETo×Kc
where *ETc* is the crop evapotranspiration (mm day^−1^); *Kc* is the crop coefficient, adjusted based on the data of relative humidity and wind velocity measured at a 2-m height of the study area; and *ETo* is the reference evapotranspiration (mm day^−1^) that was estimated using FAO CROPWAT software ver. 8. This software uses the modified FAO Penman–Monteith equation to estimate the ETo, as reported by Allen et al. [49]. The daily climatic parameters collected from the meteorological station closest to the research site were applied to the modified FAO Penman–Monteith equation to estimate the ETo. The detailed climatic parameters of two growing seasons are given in Table 1. Based on the above calculation, the total volumes of irrigation water for 100% ETc treatment were approximately 595.6 and 623.1 mm ha^−1^ in the first and second growing seasons, respectively. This irrigation water volumes were reduced to 25% and 50% for the 75% and 50% ETc treatments, respectively.

### 2.4. Measurements

#### 2.4.1. Thermal Measurements

Using a handheld infrared thermal camera (Ti-32, Fluke Thermography, Glottertal, Germany) between 10:00 and 13:00 h, infrared thermal images (IR images) of each soybean plot were captured at the beginning bloom (R1) and full seed (R6) growth stages. The camera was operated in a wavelength range of 7.5–14 µm, which is regarded as optimal for imaging applications that use heat signatures [50]. The camera had a ±0.2 °C accuracy in the temperature range of −20 to 600 °C. It was equipped with a 320 × 240 pixel micro-bolometer and a standard lens field of view of 23° × 17°. The thermal sensitivity of the camera was ≤0.05 °C at 30 °C with a spatial resolution of 1.25 mRad and minimum focus distance of 15 cm. The emissivity for measurements of plant canopy was set to 0.95 for the wet reference and the target leaves and to 0.96 for the dry reference [51]. The camera was kept at a height of approximately 0.80 m above the plant canopy at both growth stages. The IR image obtained for each plot was analyzed using the SmartView Fluke IR imaging software (version 3.2, Fluke Corporation, Plymouth, MN, USA) for data extraction and image visual. For temperature analysis of the individual leaves within the canopy, a polygon area was fitted around the leaf, and the leaf temperature was calculated as a mean value for the polygon area by the software. From each image, the average leaf temperature of the ten leaves was calculated. The data of the infrared thermal images were applied using the following equations to calculate the CWSI [52] and NRCT [19]:(2)$$CWSI=\frac{\left[\left(T_{c}-T_{a}\right)c-\left(T_{c}-T_{a}\right)l\right]}{\left[\left(T_{c}-T_{a}\right)u-\left(T_{c}-T_{a}\right)l\right]}$$
where (*T_c_* − *T_a_*) is the difference between canopy (*T_c_*) and air (*T_a_*) temperature (°C) for the current condition. The upper boundary temperature (*T_c_* − *T_a_*)*u* and lower boundary temperature (*T_c_* − *T_a_*)*l* represent non-transpiring and full transpiring conditions, respectively. When the crop was fully watered, the value of *CWSI* was close to 0, whereas for a crop under high water-stress conditions, the value of *CWSI* was close to 1.
(3)$$NRCT=\frac{T-T_{min}}{T_{max}-T_{min}}$$
where *T* is the real infrared temperature measured in the canopy, *T_min_* is the lowest temperature measured in the whole-field trial, and *T_max_* is the highest temperature in the whole-field trial. This method has the advantage of only requiring measurement of infrared temperatures.

#### 2.4.2. Spectral Reflectance Measurements

Spectral reflectance data of soybean canopies were also collected at the R1 and R6 growth stages under windless and sunny conditions within ±2 h of solar noon. The diffuse reflection characteristics of canopies were captured using a passive bidirectional reflectance sensor system (Handy Spec Field^®^, tec5, Oberursel, Germany). This device has two units of a Zeiss MMS1 silicon diode array spectrometer, one connected to a diffuser to detect solar radiation as a reference signal, while the second unit simultaneously captures the spectral signatures reflected from the canopy at a spectral range from 302 to 1148 nm with a spectral resolution of approximately 3.3-nm bandwidth and a field of view of 12°. The sensor analyses the reflected radiation in 256 spectral channels [45]. The spectral reflectance from the canopy was taken at the nadir looking angle of 25°, at approximately 0.80 m above the crop canopy, to cover a field view with a diameter of approximately 23 cm. A polytetrafluoroethylene (PTFE) white standard with approximately 100% reflectance was used to calibrate and optimize the canopy reflectance measurements. The calibrations were done before and after the measurements of canopy spectral reflectance for each plot. To acquire an accurate value of spectral reflectance, the average of four sequential measurements and ten scans for each was considered as one spectrum measurement for each plot. Finally, the spectrometer software program was used for preprocessing spectral reflectance of each plot at 1-nm intervals prior to calculation of the published or new SRIs.

#### 2.4.3. Plant Trait Measurements

After the spectral reflectance data had been collected, an area of 1.3 m length (1.0 m^2^ and approximately 20 plants) in the middle two lateral drip lines from each plot was cut from the ground level, and its biomass fresh weight (*BFW*) was immediately recorded. The plant samples were cut into small pieces, oven-dried at 80 °C to a constant weight, and weighed to record the biomass dry weight (*BDW*). The data of BFW and BDW were applied to the following equation to determine the canopy water mass (*CWM*) [45]:(4)$$CWM=\frac{BFW-BDW}{A}$$
where *A* is the area of harvested biomass for each treatment.

When the plants reached the maturity stage (R8), an area of 10.0 m length (7.5 m^2^) in the middle two lateral drip lines of each plot was harvested by hand, sun-dried for one week, and manually threshed to separate the seeds. The final SY was then weighed and expressed as Mg ha^−1^ based on the harvested area.

### 2.5. Selection of Published and Newly Constructed Spectral Reflectance Indices

Based on the spectral reflectance data, 11 SRIs (seven published SRIs and four new SRIs constructed in this study) were selected to examine their efficiency for assessing the different plant traits. These SRIs are listed with their equations and references in Table 2. Additionally, the SRIs were formulated based on the VIS and NIR of the electromagnetic spectrum sensitive to changes in photosynthetic efficiency, leaf cellular structure, leaf chlorophyll and other pigment content, and plant water content.

The new SRIs were constructed based on 2-D correlogram maps established using the pooled data of replications, irrigation regimes, and seasons of the spectral reflectance data measured at the R6 growth stage for each plant trait (Figure 2). These 2-D correlogram maps show the coefficients of determination (R^2^) for the relationships between values of plant traits and the SRIs calculated from all possible combinations of dual wavelengths of binary in the entire spectrum range (302–1148 nm). The hotspot regions for the best R^2^ determine the best relationships of SRIs with plant traits. Based on hotspots of the best R^2^, the new SRIs were based on the combined information from photosynthetic activity, leaf pigmentation, and biomass accumulation in the VIS (531, 550, 560, 580, 610, 640, 660, and 670 nm) and red-edge (715 nm) regions and the weak absorption bands located in the NIR region (800, 850, 890, 900, 970, 1070, and 1100 nm). The lattice package in R statistics v.3.0.2 (R Foundation for Statistical Computing 2013) was employed to create the different 2-D correlogram maps from the spectral reflectance data measured at the R6 growth stage.

### 2.6. Statistical Analysis

The data obtained for the different measurements (plant traits, infrared thermal criteria, and SRIs) were subjected to an analysis of variance (ANOVA) appropriate for a randomized complete block design to assess any significant differences in these measurements between the three irrigations regimes. Furthermore, the mean differences for the different measurements among the irrigation regimes were compared using Fisher’s least significant difference (LSD) test at a *p* ≤ 0.05 significance level. A combined ANOVA was performed to analyze the differences among irrigation regimes across the two growing seasons using Bartlett and Shapiro–Wilk tests. The combined analysis indicated homogenous variances across seasons for the different measurements, and therefore, the data of the two seasons were combined. Irrigations regimes and seasons were considered a fixed effect, while the replicate was considered a random effect. The statistical package R software version 3.6.1 (R Core Team 2020) was used for this analysis.

Simple linear regression analysis was performed to establish the relationships of different plant traits with thermal infrared criteria and SRIs under each irrigation regime across growth stages (*n* = 24), at each growth stage across irrigation regimes (*n* = 16), and under each irrigation regime at specific growth stages (*n* = 8). The significance level of R^2^ values for these relationships was *p* ≤ 0.05.

## 3. Results and Discussion

### 3.1. Response of Growth Performance and Yield to Irrigation Regimes at Different Growth Stages

Based on ANOVA, the SY in two growing seasons as well as the other plant traits (BFW, BDW, and CWM), measured at the R1 and R6 growth stages across the two seasons, presented significant differences (*p* < 0.05) between the three irrigation regimes (Table 3). Averaged over the two seasons, applying 25% (75% ETc) and 50% (50% ETc) less water requirement resulted in SY decreases of 21.9% and 48.9%, respectively, when compared with the 100% ETc treatment. Fisher’s protected LSD test showed that the differences in BFW, BDW, and CWM between the 100% and 75% ETc treatments were not significant, with 50% ETc producing the lowest values for these traits at the R1 growth stage, whereas at the R6 growth stage, these traits for the irrigation regimes were ranked 100% ETc > 75% ETc > 50% ETc. Generally, the 75% ETc treatment decreased BFW, BDW, and CWM by 16.5%, 7.8%, and 18.8% at the R1 growth stage and by 32.4%, 18.4%, and 38.5% at the R6 growth stage, respectively, when compared with the 100% ETc treatment. The 50% ETc treatment decreased BFW, BDW, and CWM by 35.2%, 17.1%, and 61.2% at the R1 growth stage and by 61.2%, 40.2%, and 70.3% at the R6 growth stage, respectively, when compared with the full irrigation treatment (100% ETc) (Table 3). These results indicate that the negative effects of moisture stress on growth performance and yield of soybean depend not only on the intensity of moisture stress but also on the sensitivity of the phenological growth stage to the intensity of the moisture stress. Several studies have reported that most soybean phenological growth stages are very sensitive to soil moisture stress but with varying degrees of response. In general, moisture stress during the vegetative growth stage (emergence to seed development) had negative effects on growth performance in terms of total dry matter accumulation (TDM), crop growth rate, leaf area expansion, LAI, internode length, and plant height. This negative effect of moisture stress during this growth stage resulted in a significant reduction in the elongation and expansion of plant cells, photosynthetic rate, and nitrogen fixation as well as induced rapid leaf senescence and early maturity [13,16,58,59,60]. Board and Kahlon [61] reported that moisture-stress treatments begin showing diminished TDM and LAI by the late vegetative or early reproductive growth stages. However, based on the response of the different yield components to soil moisture stress, the majority of studies have conclusively demonstrated that the growth stages from R1 to R6 have been identified as being the most sensitive growth period to soil moisture deficits in soybean. Consequently, the soil moisture-stress sensitivity of the R6–R7 period (rapid seed filling) is less than half that of the R1–R6 period; therefore, SY loss is generally twice as great for the R1–R6 than the R6–R7 period [61,62]. Their report confirms the finding of this study that the irrigation treatments showed significant differences in BFW, BDW, and CWM at both the R1 and R6 growth stages (Table 3).

Importantly, the results of this study further confirm that these plant traits could serve as benchmark indicators for soil moisture stress at critical growth stages of soybean because they are closely associated with several morpho-physiological properties of soybean under drought stress such as canopy wilt, photosynthesis efficiency, leaf expansion and elongation, flower and pod abortion, plant water status, and other growth and developmental traits [13,16,58,61,63,64,65,66,67]. For example, Board and Kahlon [61] mentioned that, if we know the TDM at the R1 and seed-development (R5) growth stages, we could predict early which potential soil moisture-stress levels cause significant soybean yield loss. Thus, simultaneous and frequent monitoring and assessment of these plant traits in a quick, easy, and nondestructive manner could play a vital role in managing deficit irrigation and in optimizing crop production by planning an optimal irrigation schedule (timing and volume), which is presented and explained in the following sections.

### 3.2. Thermal Canopy Temperature-Based Criteria and Performance in Assessment of Vegetative Growth Traits and Seed Yield

The earliest response to soil moisture stress is stomatal closure, and there is a tendency for stomata to close with decreasing soil moisture to avoid excessive water loss through transpiration. However, stomatal closure for an extended period results in a considerable rise in leaf temperature because of the correlation between transpirational cooling ability and leaf surface temperature in which the latent heat of vaporization plays a vital role in cooling the leaf surface through the conversion of water to water vapor [24,25,68,69]. Therefore, thermal canopy temperature-based criteria such as CWSI, which is based on the divergence between the upper and lower boundaries of canopy-to-ambient air temperature difference [52], and NRCT, that is based on the actual infrared temperature of the canopy [30], could serve as quick, easy, and nondestructive guides for understanding and monitoring the response of plant phenotyping traits to soil moisture stress and can thus be used as irrigation scheduling tools [17,50,70,71,72,73]. This study illustrated significant differences (*p* < 0.05) in the CWSI and NRCT among the three irrigation regimes at both the R1 and R6 growth stages. Based on Fisher’s protected LSD test, the values of CWSI and NRCT for the irrigation regimes were ranked 100% ETc < 75% ETc < 50% ETc at both growth stages, which is the inverse ranking of the vegetative growth traits and SY (Table 3). These results indicate that increasing soil moisture stress could lead to an imbalance between the water absorbed by the plants and that required to cool the canopy through the transpiration process and thus ultimately to increased CWSI and NRCT values; the opposite was true for the well-watered conditions. The resulting significant differences in CWSI and NRCT among the irrigation regimes indicate that both of the thermal canopy temperature-based criteria can be exploited to determine irrigation thresholds and to avoid yield loss through effective management of irrigation schedules. Similarly, Irmak et al. [74] found that CWSI is a useful thermal canopy criterion to monitor and quantify water stress in maize under Mediterranean semiarid climatic conditions, in which the value of CWSI should remain below 0.22 to avoid yield loss. Steele et al. [75] also reported no significant reduction in maize yield when irrigation scheduling was based on the CWSI threshold of 0.2; a CWSI value of above 0.6 has been reported to reduce grain yield. Using the CWSI as a guide tool for irrigation scheduling, O’Shaughnessy et al. [76] successfully identified an area within a field where a decrease in soil moisture resulted in a significant reduction in water use and yield of soybean and cotton under various irrigation regimes using empirical CWSI. The NRCT also showed a significant negative relationship with grain yield of different barley cultivars subjected to mild and severe water stress [19]. The findings of these studies and our results confirm that thermal canopy temperature-based criteria could provide important insights for the effective management of deficit irrigation. Additionally, they can be used as a rapid and nondestructive tool to track the changes in several plant phenotyping traits for the early detection of water stress to avoid yield loss.

Table 4 shows the best models of the regressions and coefficients of determination (R^2^) for the relationships between both criteria of thermal canopy temperature (CWSI and NRCT) and the vegetative growth traits and SY at each growth stage across three irrigation regimes, for each irrigation regime across two growth stages, as well as for each irrigation regime at each growth stage. Both thermal canopy temperature criteria showed strongest relationships with the three plant traits and SY (R^2^ ranged from 0.67 to 0.94) when the data derived from different irrigation regimes were pooled at each growth stage (*n* = 24), and the linear equation best modeled these relationships. When the data derived from different growth stages were pooled for each irrigation regime (*n* = 16), both thermal temperature criteria still exhibited moderate to strong relationships with the three plant traits (R^2^ ranged from 0.54 to 0.84), but they failed to exhibit any relationship with SY. The linear equation best modeled these relationships with a few quadratic polynomial relationships (Table 4). When the relationships of both thermal temperature criteria with plant traits and SY were analyzed for each irrigation regime at each growth stage (*n* = 8), the relationship of CWSI with plant traits and SY were weak or insignificant except for SY under 50% ETc at R6, which showed the strongest linear relationship with CWSI (R^2^ = 0.80).

Additionally, the CWSI failed to exhibit a relationship with any traits under 75% ETc at both growth stages whereas the NRCT exhibited a weak to strong relationships with BFW (R^2^ = 0.30–0.85), CWM (R^2^ = 0.27–0.83), and SY (R^2^ = 0.35–0.80) when the data were analyzed under each irrigation regime at each specific growth stage. The NRCT had a strong relationship with BDW at both the R1 (R^2^ = 0.90) and R6 (R^2^ = 0.96) growth stages only under 50% ETc (Table 4). In accordance with these findings, we can indicate that the thermal canopy temperature-based criteria could be effectively used to assess and track the changes in plant water status, biomass production, and SY of soybean at the early reproductive growth stages (between R1 and R6). Very similar results have been obtained by Kumar et al. [77]; they also found that canopy temperature depression measured at the early reproductive stage (approximately 21 and 35 days after 50% flowering) explained a major proportion of the variation in SY of soybean genotypes under both full and limited irrigation regimes.

A similar association was also found at the anthesis growth stage and closely after in bread wheat grown under dry land conditions [78]. Additionally, the results of this study reveal that the thermal canopy temperature-based criteria also efficiently explained the variability in plant water status in terms of BFW, CWM, and BDW. However, they failed to explain the variability in SY under each irrigation regime separately when the data of the two growth stages were combined for each irrigation regime. This failure of thermal canopy temperature-based criteria to detect the variability in SY could be attributed to the differences in canopy development, environmental conditions, canopy coverage of the ground, green leaf area, and transpiration efficiency between the two growth stages that may be sufficient to blur the effectiveness of thermal criteria in detecting SY variability when the data of both growth stages were combined for each irrigation regime. This finding indicates that the suitability of thermal canopy temperature-based criteria for early detection of SY variability under different irrigation regimes must be separately identified based on the different growth stages. Han et al. [73] also reported that the theoretical CWSI, which takes climatic variables into account, provides a better explanation of the variability in maize grain yield between growth stages and irrigation regimes than the empirical CWSI. The results of the relationships between thermal criteria and SY at each growth stage separately also confirmed this statement and found that the NRCT at the R1 growth stage and both thermal criteria at the R6 growth stage showed moderate to strong relationships with SY under each irrigation regime (R^2^ ranged from 0.35 to 0.80) (Table 4). Additionally, the quadratic polynomial regression was the best model fit to these relationships (Table 4). This indicates that the efficiency of different thermal canopy temperature-based criteria in detecting SY might depend on the degree of change in transpirational cooling of the canopy and water uptake under different irrigation regimes and climatic variables at each growth stage.

### 3.3. Canopy Spectral Reflectance and Performance for Assessment of Vegetative Growth Traits and Seed Yield

Fortunately, when plants are subjected to soil moisture stress, there can be substantial variations in several internal biochemical and biophysical plant characteristics, such as internal structural characteristics of the leaf mesophyll, leaf dry matter content, leaf pigment contents, and leaf water content, which eventually induce substantial changes in the spectral signatures that are reflected from the plant canopy in the VIS, NIR, and SWIR regions of the electromagnetic spectrum [6,8,36,38,79,80,81]. Consequently, these changes in the spectral reflectance of the canopy have been exploited for indirect assessment of different plant phenotyping traits that are eventually directly related to changes in soil water content. Therefore, to exploit the spectral reflectance data for indirect assessment of plant traits of interest, the spectral reflectance values at specific and effective wavelengths between the VIS and SWIR domains were used to create specific SRIs. Ultimately, these SRIs have been used to assess different plant traits indirectly. In this study, the sensitive wavelengths that are used to choose the published SRIs or to construct the new ones were identified through 2-D correlogram maps that show the hotspot regions of the best R^2^ and to determine the best relationships between the values of soybean plant traits and all possible dual wavelength combinations of binary in the spectral range 302–1148 nm as ratio SRIs (Figure 2). In general, depending on the plant traits and SY, the hotspot regions of the best R^2^ were located at the VIS region on the horizontal axis with VIS or NIR regions on the vertical axis and at the NIR region on the horizontal axis with VIS or NIR regions on the vertical axis (Figure 2). These results indicate that the SRIs that incorporate wavelengths from both VIS and NIR could be effectively used for the indirect assessment of plant water status, biomass production, and SY of soybean under a wide range of irrigation regimes. Because the water bands in the NIR region can penetrate deeper into the canopy and are more sensitive to changes in the internal leaf structure, the SRIs incorporating the NIR wavelengths were informative in assessing the variability in growth, water status, and production of various field crops under different irrigation treatments [17,20,30,44,45,82,83]. Because the wavelengths in the VIS region are dependent on pigment content and photosynthetic capacity, the SRIs incorporating the VIS wavelengths were found to be effective in reflecting the status of growth and production of crops under different environmental conditions [8,46,84,85,86]. Although no wavelengths in the VIS region are directly related to plant water status, some SRIs incorporating the VIS wavelengths were found to be sensitive to plant water status [20,21,82,87]. This is likely because the loss of cell turgor under soil moisture stress leads to a decrease in cell volume, which eventually results in chlorophyll degradation, pigment photo-oxidation, and insufficient synthesis of chlorophyll [88], thereby influencing the spectral reflectance not only in the NIR but also in the VIS regions. Additionally, the wavelengths in the red-edge region (700–780 nm) carry important information on biomass quantity, and thus, the SRIs incorporating the red-edge wavelengths were found to be effective for tracking changes in biomass and SY under different environmental conditions [20,41]. These findings indicate that the SRIs based on combined information from the VIS, red-edge, and NIR regions could be used to assess and predict several plant phenotypic traits related to growth, water status, and crop production under a wide range of irrigation regimes. For example, the normalized difference vegetation index (NDVI), which incorporates wavelengths from the red (620–690 nm) and NIR (760–900 nm) regions, has been shown to correlate well with different plant phenotypic traits such as leaf water content, leaf water potential, aboveground biomass, LAI, ground cover, and final yield in different field crops under either normal or stress conditions [8,19,20,21,36,38,46,47]. The photochemical reflectance index (PRI) that incorporates the major green wavelengths (531 and 570 nm) has also been found to be important in estimating several plant phenotypic traits in various environments [46,89]. The water index (WI) and different normalized water indices (NWIs) that focus on the weak water absorption bands within the NIR region (near 970 nm) are appropriate to detect crop production as well as the plant phenotypic traits that are associated with plant water status under soil moisture-stress conditions [17,19,36,90,91].

In this study, all tested SRIs that are based on green, red, red-edge, and NIR spectra showed significant variation (*p* < 0.05) between the three irrigation regimes at the R1 and R6 growth stages across two seasons (Table 3). Additionally, the values of PRI_(531,570)_, SRI_(610,580)_, and SRI_(660,560)_, which are based on red and green wavelengths, as well as the values of SRI_(678,1070)_ and SRI_(800,970)_, which are based on red and NIR wavelengths, decreased from the R1 to the R6 growth stage and showed a continuous increase from the 100% ETc to the 50% ETc treatments. Contrarily, the values of NDVI_(800,640)_, SRI_(890,715)_, DWI_(970,670)_, DWI_(1100,670)_, and NWI-2_(970,850)_, which are based on red, red-edge, and NIR wavelengths, increased from the R1 to the R6 growth stages, but they showed a continuous decrease from the 100% ETc to the 50% ETc treatments except for NWI-2, which increased from the 100% ETc to the 50% ETc treatment (Table 3). These findings indicate a substantial increase in the canopy spectral reflectance of soybean in the VIS region of the spectrum, especially in the red region, due to soil moisture stress. Wijewardana et al. [8] similarly found that the soil moisture stress at 60%, 40%, and 20% ETc led to a substantial increase in soybean canopy spectral reflectance in the VIS region as compared to 80% and 100% ETc. Additionally, the reduction in the SRIs based on green and red wavelengths from the R1 to the R6 growth stages indicates a lower pigment content and progressive senescence of photosynthetic organs as the growth stage progressed from R1 to R6 and therefore, increased spectral reflectance in the VIS region. Furthermore, because the NIR-based SRIs are based on spectral reflectance at the weak water absorption bands (near 970 and 1200 nm) and on spectral reflectance at 850 nm and 900 nm caused by the multiple scattering and reflection of radiation by different leaf tissues, the values of these SRIs showed a continuous decrease from the 100% ETc to the 50% ETc treatment due to a greater decrease in the canopy spectral reflectance in the NIR region for the soil moisture-stress treatments. Additionally, because the canopy contains a higher total quantity of biomass at the R6 stage that is expressed in this study as BDW (Table 3), the values of the NIR-based SRIs showed a continuous increase from the R1 to the R6 growth stages due to a greater increase in the canopy spectral reflectance in the NIR region of the spectrum at the R6 than the R1 growth stage (Table 3). Prasad et al. [90] also found a similar trend of increasing values of NIR-based SRIs such as WI and NWI with the advancement of growth stages in winter wheat under rainfed conditions.

Previous studies have reported that specific growth stages and conditions at the time of measuring the canopy spectral reflectance can influence the performance of SRIs in the assessment of plant phenotypic traits [6,81,90,92]. For example, some studies report the early stage of durum wheat (anthesis stage) as the best growth stage for assessing several plant phenotypic traits for both rainfed and supplemental irrigation treatments using SRIs. However, in other studies, the late-stage (grain-filling stage) showed the strongest relationship between SRIs and plant phenotypic traits under well-watered conditions. Additionally, based on specific growth stages, some SRIs performed best for assessing plant phenotypic traits under limited irrigation regimes but failed to assess these traits under full irrigation regimes and vice versa. Furthermore, various SRIs failed to assess plant phenotypic traits within each water regime, whereas they successfully assessed them across different irrigation regimes. All of this evidence reveals that growth stage and crop growth conditions may play a vital role in the efficiency of assessment and monitoring of different plant phenotyping traits using SRIs, especially when the canopy spectral reflectance is captured with a wide range in both variables.

Regarding the relationships between SRIs and plant traits, and SY within each growth stage across irrigation regimes (*n* = 24) and across two growth stages and irrigation regimes (*n* = 48), the results of this study showed that the majority of the SRIs examined showed strong relationships with BFW (R^2^ = 0.78–0.84) and CWM (R^2^ = 0.78–0.85) and moderate relationships with BDW (R^2^ = 0.51–0.63) and SY (R^2^ = 0.59–0.67) at the R1 stage whereas these SRIs showed moderate to strong relationships with the four traits (R^2^ = 0.54–0.85) at the R6 stage (Figure 3). Furthermore, the water-based NIR indices (WI_(900,970)_, SRI_(800,970)_, and NWI-2_(970,850)_) exhibited a weak relationship with the four traits at the R1 stage (R^2^ = 0.29–0.45) and a moderate relationship at the R6 stage (R^2^ = 0.52–0.60) whereas they failed to assess the four traits when the SRI data of the growth stages and irrigation regimes were combined (Figure 3). Although most SRIs exhibited moderate to strong relationships with the four traits within each growth stage, these SRIs showed a weak relationship with SY (R^2^ = 0.16–0.47) and showed moderate to strong relationships with BFW, CWM, and BDW (R^2^ = 0.55–0.89) when the SRI data of the growth stages and irrigation regimes were combined (Figure 3). These results indicate that, in using SRIs, the specific growth stages must be taken into account when assessing the plant phenotyping traits, especially the aboveground biomass and SY. In this study, the SRIs, especially the water-based NIR indices, were more effective in estimating the BDW and SY of soybean at the R6 stage than at the R1 growth stage (Figure 3). The primary reason for this may be that the biomass saturation and high LAI at the R1 stage could make the SRIs less efficient in assessing BDW and SY than at the R6 stage. Generally, the commonly used SRIs are usually saturated at LAI > 3 and become effective for assessing plant phenotyping traits when the values of LAI decrease to approximately 2 [81,90,92,93]. Additionally, lowered pigment content and progressive senescence of photosynthetic organs as growth progresses toward the reproductive stage may explain why the relationship of SRIs with BDW and SY becomes stronger at the R6 than the R1 growth stage. However, the greater sensitivity of BFW and CWM to plant water content than to LAI may explain why the relationships between both traits and most SRIs were strong and similar for each growth stage and across all data.

Figure 4 shows that all the SRIs examined failed to assess SY when the data of two growth stages were combined for each irrigation regime (*n* = 16). All the SRIs examined except WI_(900,970)_, SRI_(800,970)_, and NWI-2_(970,850)_ showed strong relationships with BFW, CWM, and BDW under 100% ETc (R^2^ = 0.70–0.96) and moderate to strong relationships with these three traits under 75% ETc (R^2^ = 0.55–0.88). Under 50% ETc, these SRIs still showed moderate to strong relationships with BFW (R^2^ = 0.60–0.89) and BDW (R^2^ = 0.56–0.85) but showed weak relationships with CWM (R^2^ = 0.32–0.48). The three excluded SRIs (WI_(900,970)_, SRI_(800,970)_, and NWI-2_(970,850)_) exhibited a weak relationship (R^2^ = 0.26–0.35) and weak to moderate relationships (R^2^ = 0.42–0.61) with the three traits under the 100% ETc and 75% ETc treatments, respectively, whereas, they exhibited moderate to strong relationships (R^2^ = 0.55–0.87) with BFW and BDW under 50% ETc (Figure 4). Interestingly, although all SRIs failed to assess SY within each irrigation regime across two growth stages, they exhibited moderate relationships with SY under 100% ETc at the R6 stage (R^2^ = 0.34–0.53), weak relationships under 50% ETc at the R1 stage (R^2^ = 0.31–0.47), and moderate to strong relationships under 75% ETc at the R1 stage (R^2^ = 0.46–0.71) (Table 5). Additionally, the R^2^ values of the relationships between SRIs and BFW, BDW, and CWM were dramatically decreased when the relationships were analyzed for each irrigation rate at specific growth stages (Table 5).

These findings indicate that, when the relationships between SRIs and plant phenotyping traits are obtained across different growth stages for each irrigation regime separately or for each irrigation regime at a specific growth stage, they must be used cautiously. This precaution is due to random changes in the relative performance of each plant phenotyping trait from one growth stage to another and one irrigation regime to another, which eventually influences the canopy spectral reflectance patterns and its effectiveness for the indirect assessment of these traits. This study found that the SRIs were much more effective for indirectly estimating vegetative growth traits such as BFW, BDW, and CWM when the data of the SRIs were combined across different growth stages for each irrigation regime separately or vice versa. However, rather than combining SRIs across different growth stages for each irrigation regime, the SY should be assessed for each irrigation regime separately at specific growth stages. The primary reason for this may be that the spectral reflectance measurements of the combined full irrigation regime (100% ETc) and the early growth stage (R1) may create problems related to saturation of SRIs due to the increase of LAI and high canopy densities, at which most SRIs are usually saturated when the LAI is larger than 3, whereas when the spectral reflectance measurements of severe moisture stress (50% ETc) and late growth stage (R6) are combined, it may create problems related to soil background reflectance due to the decrease in LAI and to the exposure of bare soil. Consequently, once the problems related to saturation or soil background reflectance were removed, the assessment of SY using SRIs was improved. The results presented in Table 5 also fully confirm this observation and show that, under 100% ETc, the SRIs were much more effective for estimating SY at R6 than at R1; the opposite was true under 75% ETc and 50% ETc. Similarly, previous studies have reported that most SRIs measured at the early growth stage performed best for estimating grain yield under severe moisture stress (with a LAI of approximately 2) but failed to assess grain yield under well-watered conditions whereas the relationship between SRIs and grain yield under well-watered conditions became stronger as the growth stage progressed toward grain filling [90,92,93,94]. Christenson et al. [46] also reported that no trends were observed for the accurate estimation of SY using wavebands and SRI models across different irrigation regimes in soybean.

Contrary to SY, rather than estimating the vegetative growth traits (BFW, BDW, and CWM) based on the SRIs of each irrigation regime at specific growth stages, these traits should be assessed for each irrigation regime across two growth stages or vice versa (compare R^2^ values for Figure 3, Figure 4 and Table 5). The reason may be that, because these traits provide direct information on plant water status, pigment content, and biomass accumulation, the SRIs could be limited in accurately estimating these traits unless the treatments cause noticeable alterations in the internal plant water status, leaf structure, and pigment content. This observation indicates that the efficiency of SRIs for estimating vegetative growth traits (BFW, BDW, and CWM) may depend on a combination of the magnitude of the effects of irrigation regimes and growth stages. Previous studies have also reported that several SRIs did not exhibit relationships with several physiological traits until the data of different irrigation regimes were combined [6,20,36,47,90].

## 4. Conclusions

In this study, we investigated the potential use of thermal imaging criteria and SRIs for assessing the growth performance and production of drip-irrigated soybean exposed to different irrigation regimes for two years at two growth stages. The results indicated that there were significant differences (*p* < 0.05) among different vegetative growth traits, thermal imaging criteria, and SRIs between three irrigation regimes at two growth stages. The values of both thermal imaging criteria and SRIs based on red, red-edge, and NIR wavelengths increased from the R1 to the R6 growth stage. Additionally, the thermal imaging criteria showed a continuous increase from the 100% ETc to the 50% ETc treatment; the opposite was true for SRIs. The values of SRIs based on red and green wavelengths or red and NIR wavelengths decreased from the R1 to the R6 growth stages and showed a continuous increase from the 100% ETc to the 50% ETc treatments. The efficiency of thermal imaging criteria and SRIs in estimating the different vegetative growth traits and SY depend on the analyses of data for each irrigation regime and growth stage specifically or the combined data of growth stages for each irrigation regime or vice versa. Finally, this study demonstrated that thermal imaging criteria and SRIs could be used as rapid, easy, cost effective, and nondestructive tools for assisting soybean growers in making informed field management decisions regarding irrigation to reduce the negative effects of deficit irrigation on the growth and production of soybean.

## Figures and Tables

**Figure 1 sensors-20-06569-f001:**
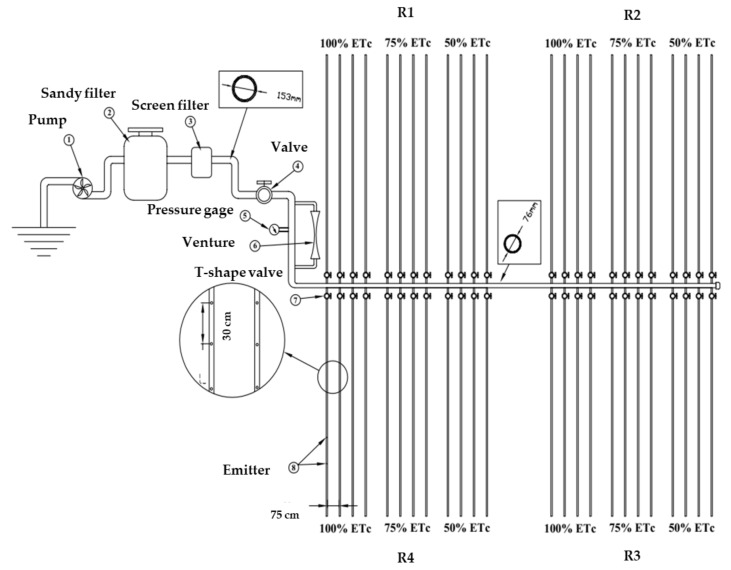
Schematic diagram of the experimental layout that includes three irrigation regimes and four replications showing locations of irrigation regimes and replications.

**Figure 2 sensors-20-06569-f002:**
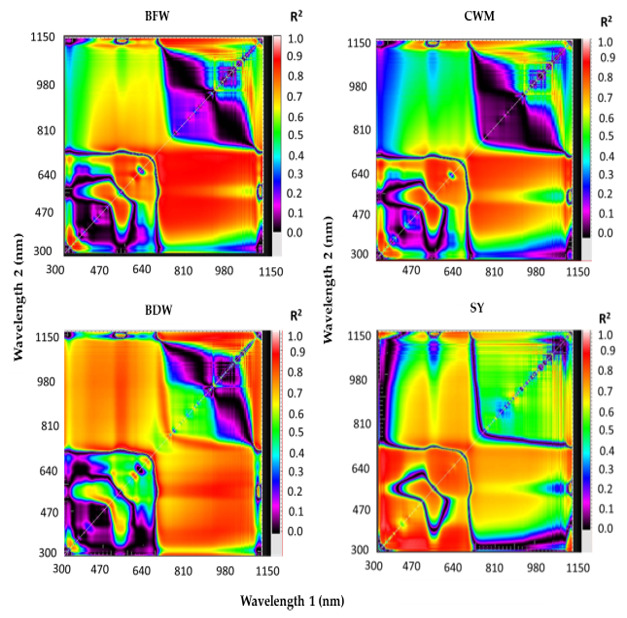
Two-dimensional correlograms show the coefficients of determination (R^2^) for the relationships between values of vegetative growth traits (biomass fresh weight (BFW), biomass dry weight (BDW), canopy water mass (CWM), and seed yield (SY)) and the spectral reflectance indices calculated from all possible combinations of dual wavelengths of binary in the entire spectrum range (from 302 to 1148 nm) using the pooled data of replications, irrigation regimes, and seasons at the full seed (R6) growth stage.

**Figure 3 sensors-20-06569-f003:**
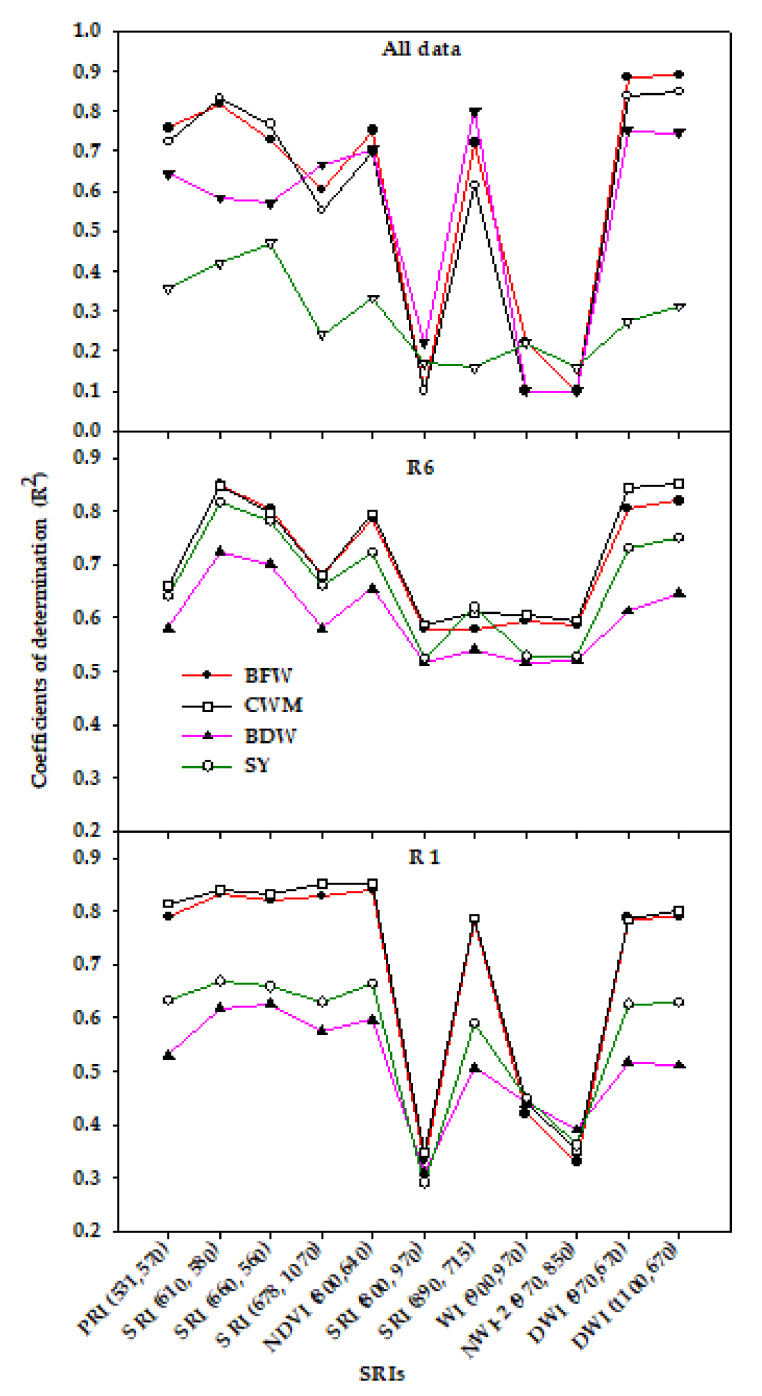
Coefficients of determination (R^2^) for the relationship between different spectral reflectance indices (SRIs) and vegetative growth traits (biomass fresh weight (BFW), biomass dry weight (BDW), canopy water mass (CWM), and seed yield (SY)) at each growth stage across three irrigation regimes (*n* = 24) and for pooled data of replications, growth stages, irrigation regimes, and seasons (*n* = 48): R1 and R6 indicate the beginning bloom and full seed growth stages, respectively.

**Figure 4 sensors-20-06569-f004:**
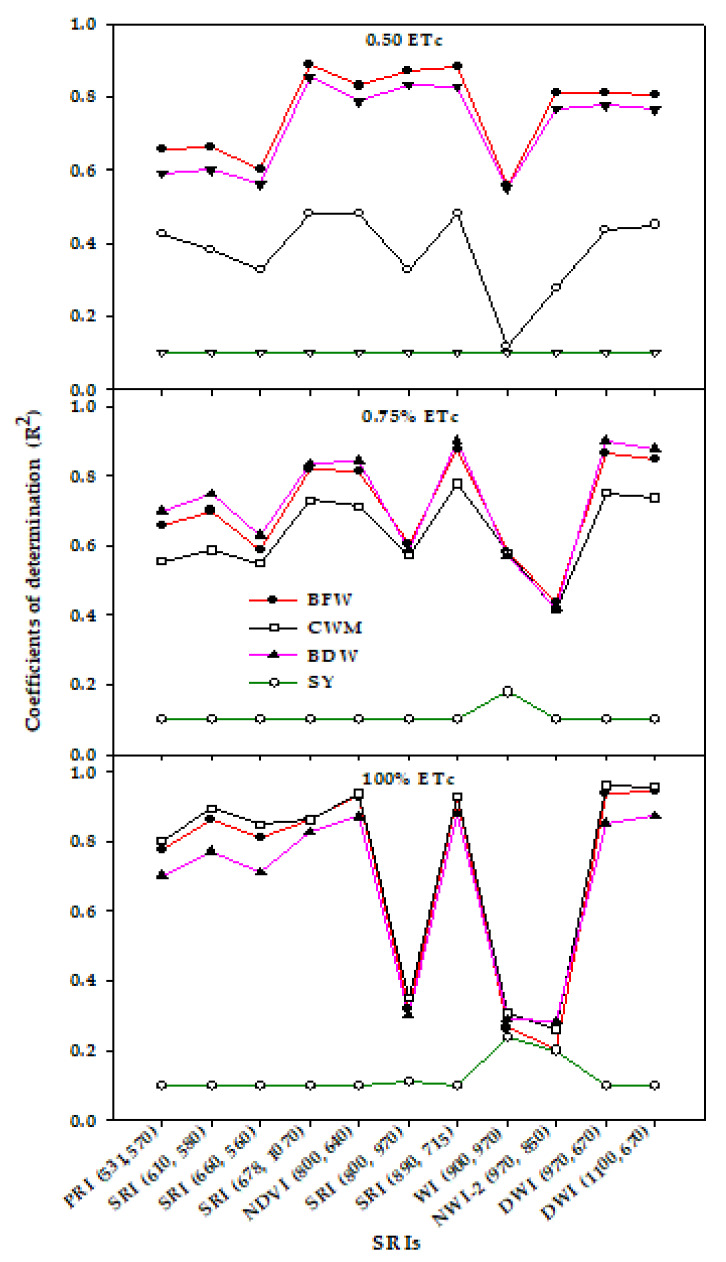
Coefficients of determination (R^2^) for the relationship between different spectral reflectance indices (SRIs) and vegetative growth traits (biomass fresh weight (BFW), biomass dry weight (BDW), canopy water mass (CWM), and seed yield (SY)) for each irrigation regimes across two growth stages (*n* = 16).

**Table 1 sensors-20-06569-t001:** Monthly agro-climatological data in the Sadat city region (30°2′41.2″ N, 31°14′8.2″ E) in the 2016 and 2017 growing seasons.

Year	Months	Temperature (°C)		Wind Speed (m s^−1^)	Relative Humidity (%)	Total Solar Radiation (MJ m^−2^ day^−1^)	Net Solar Radiation (MJ m^−2^ day^−1^)
		Maximum	Minimum				
2016	April	30.4	16.2	0.74	53.4	23.4	12.51
	May	32.1	17.0	0.81	47.2	25.6	14.71
	June	34.9	17.1	0.61	53.3	26.1	15.00
	July	35.5	22.0	0.53	62.6	22.9	14.31
	August	35.9	22.7	0.47	61.8	20.8	12.00
2017	April	32.7	17.0	0.62	51.2	24.5	14.41
	May	33.5	16.0	0.74	49.3	26.0	15.47
	June	35.9	20.8	0.65	56.4	27.6	17.14
	July	36.5	23.2	0.51	59.7	23.8	14.08
	August	35.7	23.0	0.42	63.1	22.3	13.43

**Table 2 sensors-20-06569-t002:** Full name, formula, and references of spectral reflectance indices (SRIs) tested in this study.

Spectral Reflectance Indices	Formula	References
Photochemical reflectance index (PRI, _(531,570)_)	(R_531_ − R_570_)/(R_531_ + R_570_)	[53]
Simple ratio based on 610 and 550 nm (SRI_(__610,580)_)	R_610_/R_580_	This work
Simple ratio based on 660 and 560 nm (SRI_(660,560)_)	R_660_/R_560_	This work
Simple ratio based on 678 and 1070 nm (SRI_(678,1070)_)	R_678_/R_1070_	[54]
Normalized difference vegetation index (NDVI_(800,640)_)	(R_800_ − R_640_)/(R_800_ + R_640_)	[55]
Simple ratio based on 800 and 970 nm (SRI_(800,970)_)	R_800_/R_970_	[56]
Simple ratio based on 890 and 715 nm (SRI_(890,715)_)	R_890_/R_715_	[45]
Water index (WI_(900,970)_)	R_900_/R_970_	[57]
Normalized water index 2 (NWI-2_(970,850)_)	(R_970_ − R_850_)/(R_970_ + R_850_)	[17]
Development of water index (DWI_970–670_)	R_970_/R_670_	This work
Development of Water index (DWI_1100–670_)	R_1100_/R_670_	This work

**Table 3 sensors-20-06569-t003:** Comparison of the mean values of vegetative growth traits (biomass fresh weight (BFW), biomass dry weight (BDW), and canopy water mass (CWM), seed yield (SY)), thermal canopy temperature-based criteria (crop water stress index (CWSI) and normalized relative canopy temperature (NRCT)), and eleven spectral reflectance indices among the three irrigation regimes at the beginning bloom (R1) and full seed (R6) growth stages.

	Irrigation Water Regimes					
	100% ETc	75% ETc	50% ETc	100% ETc	75% ETc	50% ETc
	2016			2017		
SY (Mg ha^−1^)	3.18a	2.45b	1.63c	3.25a	2.57b	1.654c
	**R1**			**R6**		
BFW (Mg ha^−1^)	6.11a	5.10ab	3.96b	13.32a	9.01b	5.17c
BDW (Mg ha^−1^)	1.29a	1.19ab	1.07b	4.08a	3.33b	2.44c
CWM (Mg ha^−1^)	4.83a	3.92ab	2.90b	9.24a	5.68b	2.74c
CWSI	0.18c	0.45b	0.62a	0.29c	0.61b	0.78a
NRCT	0.17c	0.42b	0.58a	0.30c	0.60b	0.79a
PRI_(531,570)_	−0.084a	−0.100b	−0.122c	−0.040a	−0.065b	−0.091c
SRI_(610,580)_	0.952c	1.00b	1.048a	0.862c	0.934b	0.999a
SRI_(660,560)_	0.731c	0.878b	1.055a	0.526c	0.712b	0.919a
SRI_(678,1070)_	0.254c	0.358b	0.501a	0.091c	0.167b	0.208a
NDVI_(800,640)_	0.800a	0.454b	0.351c	0.819a	0.665b	0.593c
SRI_(800,970)_	1.071a	1.045b	1.026c	0.991a	0.903b	0.843c
SRI_(890,715)_	1.726a	1.444b	1.325c	2.556a	2.047b	2.051b
WI_(900,970)_	1.071a	1.050b	1.029c	1.094a	1.028b	0.985c
NWI-2_(970,850)_	−0.039c	−0.026b	−0.016a	−0.022c	0.018b	0.046a
DWI_(970,670)_	4.408a	2.908b	2.150c	13.794a	6.905b	5.401b
DWI_(1100,670)_	3.230a	2.379b	1.658c	7.959a	4.354b	3.552b

Means followed by the same letter are not significantly different from one another based on Fisher’s least significant difference (LSD) test at *p* ≤ 0.05.

**Table 4 sensors-20-06569-t004:** The best models of regression and determination coefficients (R^2^) for the relationship between thermal canopy temperature-based criteria (crop water stress index (CWSI) and normalized relative canopy temperature (NRCT)) and vegetative growth traits (biomass fresh weight (BFW), biomass dry weight (BDW), canopy water mass (CWM), and seed yield (SY)) at each growth stage across three irrigation regimes (*n* = 24), for each irrigation regime across two growth stages (*n* = 16), and for each irrigation regime at each growth stage (*n* = 8): R1 and R6 indicate the beginning bloom and full seed growth stages, respectively, while L and Q indicate linear and quadratic fitting models, respectively.

Treatments		BFW		BDW		CWM		SY	
		CWSI	NRCT	CWSI	NRCT	CWSI	NRCT	CWSI	NRCT
Growth stages	R1	0.89 ^L^*	0.90 ^L^*	0.67 ^L^*	0.75 ^L^*	0.88 ^L^*	0.87 ^L^*	0.82 ^L^*	0.82 ^L^*
	R6	0.94 ^L^*	0.90 ^L^*	0.88 ^L^*	0.90 ^L^*	0.93 ^L^*	0.83 ^L^*	0.90 ^L^*	0.84 ^L^*
Irrigation water regimes	100%ETc	0.63 ^L^*	0.79 ^L^*	0.65 ^L^*	0.71 ^L^*	0.62 ^Q^*	0.82 ^L^*	0.002 ^L^	0.20 ^Q^
	75% ETc	0.72 ^L^*	0.78 ^L^*	0.72 ^L^*	0.77 ^L^*	0.84 ^Q^*	0.74 ^L^*	0.003 ^L^	0.14 ^Q^
	50% ETc	0.78 ^L^*	0.78 ^L^*	0.72 ^L^*	0.69 ^L^*	0.54 ^Q^*	0.61 ^Q^*	0.16 ^Q^	0.21 ^Q^
R1	100%ETc	0.39 ^L^*	0.30 ^Q^*	0.05 ^Q^	0.14 ^Q^	0.50 ^L^*	0.52 ^Q^*	0.01 ^L^	0.35 ^Q^*
	75%ETc	0.02 ^L^	0.36 ^Q^*	0.20 ^Q^	0.02 ^Q^	0.03 ^L^	0.27 ^Q^*	0.02 ^L^	0.53 ^Q^*
	50%ETc	0.55 ^L^*	0.38 ^L^*	0.42 ^Q^*	0.90 ^Q^*	0.13 ^Q^	0.35 ^Q^*	0.17 ^Q^	0.80 ^Q^*
R6	100%ETc	0.003 ^L^	0.85 ^Q^*	0.01 ^L^	0.19 ^Q^	0.001 ^L^	0.83 ^Q^*	0.48 ^Q^*	0.42 ^Q^*
	75% ETc	0.12 ^Q^	0.30 ^Q^*	0.20 ^Q^	0.09 ^L^	0.21 ^L^	0.43 ^Q^*	0.43 ^Q^*	0.41 ^Q^*
	50% ETc	0.05 ^L^	0.32 ^Q^*	0.49 ^L^*	0.96 ^L^*	0.52 ^Q^*	0.58 ^Q^*	0.80 ^L^*	0.61 ^L^*

* Numbers indicate statistical significance at *p* ≤ 0.05.

**Table 5 sensors-20-06569-t005:** Determination coefficients (R^2^) for the relationship between different spectral reflectance indices (SRIs) and vegetative growth traits (biomass fresh weight (BFW), biomass dry weight (BDW), canopy water mass (CWM), and seed yield (SY)) for each irrigation regime at each growth stage (*n* = 8): R1 and R6 indicate the beginning bloom and full seed growth stages, respectively.

SRIs	R1				R6			
	BFW	CWM	BDW	SY	BFW	CWM	BDW	SY
**100% ETc**								
PRI_(531,570)_	0	0.05	0.23	0.05	0.10	0.21	0.46	0.19
SRI_(610,580)_	0.03	0.13	0.13	0.12	0.49	0.65	0.26	0.34
SRI_(660,560)_	0.03	0.12	0.12	0.10	0.48	0.64	0.25	0.35
SRI_(678,1070)_	0.03	0.18	0.30	0.11	0.67	0.72	0.04	0.34
NDVI_(800,640)_	0	0.10	0.32	0.05	0.59	0.70	0.12	0.39
SRI_(800,970)_	0.17	0.49	0.27	0.14	0.45	0.56	0.15	0.45
SRI_(890,715)_	0	0.09	0.42	0.03	0.54	0.48	0.02	0.35
WI_(900,970)_	0.14	0.41	0.25	0.14	0.44	0.54	0.14	0.47
NWI-2_(970,850)_	0.16	0.46	0.27	0.12	0.46	0.56	0.14	0.46
DWI_(970,670)_	0.05	0.22	0.27	0.08	0.54	0.68	0.20	0.40
DWI_(1100,670)_	0.06	0.25	0.25	0.09	0.60	0.66	0.05	0.53
**75% ETc**								
PRI_(531,570)_	0	0	0	0.70	0.14	0.14	0.02	0.05
SRI_(610,580)_	0	0	0.01	0.48	0.25	0.30	0	0.01
SRI_(660,560)_	0.01	0	0.00	0.50	0.28	0.27	0.03	0
SRI_(678,1070)_	0.01	0	0.08	0.62	0.35	0.29	0.07	0.17
NDVI_(800,640)_	0	0.03	0.08	0.71	0.18	0.22	0	0
SRI_(800,970)_	0.02	0.01	0	0.46	0.01	0.02	0	0.16
SRI_(890,715)_	0.03	0.08	0.12	0.71	0.15	0.13	0.02	0.05
WI_(900,970)_	0	0	0	0.63	0.01	0.01	0	0.20
NWI-2_(970,850)_	0	0	0	0.54	0.02	0.02	0	0.17
DWI_(970,670)_	0.01	0	0.07	0.59	0.38	0.36	0.05	0.10
DWI_(1100,670)_	0.03	0	0.06	0.54	0.35	0.30	0.06	0.19
**50% ETc**								
PRI_(531,570)_	0.12	0.47	0.15	0.35	0.11	0.03	0	0
SRI_(610,580)_	0.28	0.31	0.27	0.37	0.08	0.03	0.01	0
SRI_(660,560)_	0.22	0.33	0.20	0.35	0.06	0.02	0.01	0
SRI_(678,1070)_	0.28	0.39	0.20	0.31	0.13	0.03	0	0
NDVI_(800,640)_	0.29	0.42	0.26	0.39	0.11	0.03	0	0
SRI_(800,970)_	0.05	0.38	0.08	0.36	0.13	0	0.23	0.02
SRI_(890,715)_	0.33	0.44	0.29	0.40	0.17	0.04	0	0.00
WI_(900,970)_	0.09	0.44	0.12	0.32	0.03	0	0.15	0.01
NWI-2_(970,850)_	0.07	0.41	0.11	0.36	0.07	0	0.20	0.02
DWI_(970,670)_	0.34	0.35	0.27	0.35	0.10	0.03	0	0.00
DWI_(1100,670)_	0.19	0.49	0.09	0.47	0.11	0.03	0	0

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
