# Peer review of "Potential of Hyperspectral and Thermal Proximal Sensing for Estimating Growth Performance and Yield of Soybean Exposed to Different Drip Irrigation Regimes Under Arid Conditions"

_sensors, 2020, doi:10.3390/s20226569_

Round 1

Reviewer 1 Report

The manuscript is well written and organized. But it still needs some improvement.

I suggested the authors re-draw all the figures. Now the figures look not professional

Figure 1. Try to keep your text upright. Especially for the items list, I didn’t see any benefit to have them rotated.

Figure 2. Clean up unnecessary content.

Figure 3. The resolution looks low. Also, adjust the fonts. Better to lay the legends on the right side.

Figure 4. Same as Figure 3. It looks coarse.

I suggest the authors add several more figures to show the experiment, equipment, or devices for data collection or measurement.

Author Response

Reviewer#1

The manuscript is well written and organized. But it still needs some improvement.

Response: We greatly appreciate your critical observations as well as your constructive and helpful comments. We hope that we could address your questions/comments by the explanations and revisions made in the manuscript. We believe that the manuscript is substantially improved after making the suggested revisions.

  • I suggested the authors re-draw all the figures. Now the figures look not professional

Response: Many thanks for this remark. All Figures have been re-drawn.

  • Figure 1. Try to keep your text upright. Especially for the items list, I didn’t see any benefit to have them rotated.

Response: Many thanks for this remark. The Figure-1 has been re-drawn.

  • Figure 2. Clean up unnecessary content.

Response: Done

  • Figure 3. The resolution looks low. Also, adjust the fonts. Better to lay the legends on the right side.

Response: The Figure-3 has been re-drawn

  • Figure 4. Same as Figure 3. It looks coarse.

Response: The Figure-4 has been re-drawn

  • I suggest the authors add several more figures to show the experiment, equipment, or devices for data collection or measurement.

Response: This figure show Handy Spec Field®, tec5, Oberursel, Germany

Reviewer 2 Report

General comments

In the manuscript, the Authors would propose a method to assessing the combination of thermal and Vis-NIR spectroscopy measurements in detecting plant growth, crop water status, and yield of soybean under arid climatic conditions at two growth stages. To do that, the Authors used thermal imaging (TI) criteria and spectral reflectance indices (SRIs)  to monitor plant water status, biomass, and seed yield (SY) of soybean exposed to 100%, 75%, and 26 50% of estimated crop evapotranspiration (ETc). Then they used a simple linear regression analysis to establish the relationships of different plant traits with thermal infrared criteria and SRIs under each irrigation regime across growth stages.

I have read the manuscript and I believe it could be a contribution of interest for the Sensors journal, but the manuscript should be shortened and be focused more on the materials and methods should be explained more especially the setup of proximal sensors and the data acquisition and the spectral and imaging processing before statistical analysis (simple linear regression) which is the weakest part in the manuscript.

I believe that the Authors have complicated their study for no reason because literature proposes more advanced and appropriate methods than what they proposed. Their methods should be supported by a good and short introduction in which the Authors explain the advantages and improvements compared to the most used methods. Optical proximal sensors combined with simple linear regression or even machine learning have been used in the literature. The Authors should justify what is new in their study and how this will advance the knowledge of sensing in precision agriculture.

They assumed that crop growth and yield only affected by water content in soil and plant, neglecting other factors affect growth and crop yield (e.g., light, heat, soil nutrients).

Finally, the Authors mixed up two types of sensing, remotes sensing, and proximal, neglecting the data preprocessing of both thermal and Vis-NIR spectroscopic measurements. The Authors used thermal and spectral data without indication of any preprocessing which is essential for spectral analysis. To my knowledge, raw spectral data is not recommended in spectral analysis especially with spectrophotometer such as tec5 that maybe have a spectral jump in the spectrum around 900 nm that required preprocessing to eliminate this jump. Furthermore, proximal sensing data required advanced data analytics- not simple linear regression- such as PLSR or other machine learning methods for better modeling.

Specific comments

The title should be further improved and made clearer, more informative, and effective. Besides, it should reflect the objectives. The confusing terms should be avoided. The authors mixed between remote sensing and proximal sensing. So, I would suggest using the title ‘’Potential of Hyperspectral and Thermal Proximal Sensing for Estimating Growth Performance and Yield of Soybean Exposed to  Different Drip Irrigation Regimes Under Arid  Conditions’’.

The abstract does not explain well what was done, what was found, and what are the main conclusions.
Keywords should be improved and made it easy for indexing and searching. There is nothing called proximal remote sensing; Proximal sensing and remote sensing are two different things. The contour map is not a well-known acronym in Proximal sensing and spectroscopic studies, use the appropriate acronym like 2-D-correlogram of R2.

The Introduction section should be improved and related to the more recent worldwide literature on the problem being investigated. The topic should be presented better and in a more general contest to allow its generalization to other areas of the world. Moreover, the Authors should briefly review better the most used methods to quantify the response of different vegetative growth traits and yield in different irrigation regimes base on optical proximal sensing and explain why they have chosen the proposed methods. Furthermore, the paper lacks a very clear and good justification for what is new and innovative about their approach.

Line 23: Hyperspectral remote sensing, should be replaced with Proximal hyperspectral sensing. You did not use remote sensing in this research.

Lines 219-221: It is not clear how you did this. Did you use any spectral data preprocessing? Have you found any jump around 950 nm (linkage point of the two detectors) or noise at both edges of the spectrum in the tec5 spectrophotometer?
Lines 242-244: Please, be specific about the setup of the calibration (interval between scanning the white reference) and how you averaged the spectra?.
Lines 316: Instead of contour map use ‘’ 2-D correlogram of the coefficients of determination…’’
Lines 337: Explain the meaning of a, b, and c in the table at the footer?
Lines 349-353: What about other factors that affect plant growth and crop yield? Please, rephrase this paragraph. 
Line 522: Use the range of wavelengths instead of bands, as you talked about specific wavelengths below, not bands.

Lines 728-732: This can be a future goal as this is not reflected in the measurement setup, the spectral data processing, as well as the advanced data analytics (e.g. machine learning and data fusion) which you did not apply in this research.

Author Response

Reviewer#2

General comments

In the manuscript, the Authors would propose a method to assessing the combination of thermal and Vis-NIR spectroscopy measurements in detecting plant growth, crop water status, and yield of soybean under arid climatic conditions at two growth stages. To do that, the Authors used thermal imaging (TI) criteria and spectral reflectance indices (SRIs) to monitor plant water status, biomass, and seed yield (SY) of soybean exposed to 100%, 75%, and 50% of estimated crop evapotranspiration (ETc). Then they used a simple linear regression analysis to establish the relationships of different plant traits with thermal infrared criteria and SRIs under each irrigation regime across growth stages.

Response: We greatly appreciate your critical observations as well as your constructive and helpful comments. We hope that we could address your questions/comments by the explanations and revisions made in the manuscript. We believe that the manuscript is substantially improved after making the suggested revisions.

  • I have read the manuscript and I believe it could be a contribution of interest for the Sensors journal, but the manuscript should be shortened and be focused more on the materials and methods should be explained more especially the setup of proximal sensors and the data acquisition and the spectral and imaging processing before statistical analysis (simple linear regression) which is the weakest part in the manuscript.

Response: Many thanks for this comment. All information related to setup of proximal sensors and the data acquisition and the thermal imaging processing before statistical analysis has been mentioned in details in the section of materials and methods under the subtitles Thermal measurements and Spectral reflectance measurements

  • I believe that the Authors have complicated their study for no reason because literature proposes more advanced and appropriate methods than what they proposed. Their methods should be supported by a good and short introduction in which the Authors explain the advantages and improvements compared to the most used methods. Optical proximal sensors combined with simple linear regression or even machine learning have been used in the literature. The Authors should justify what is new in their study and how this will advance the knowledge of sensing in precision agriculture.

Response: In the Introduction section we already explain the important of infrared thermal and proximal canopy spectral reflectance tools as a powerful alternative to traditional plant sampling techniques for the integrative assessment of multiple morpho-physiological indicators that accurately reflect crop water stress. The close relationship between the indices of both tools and plant parameters, particularly those associated with the plant water status and biomass, means that both tools can be exploited to effectively manage irrigation scheduling that involves deciding how much and when irrigation water should be applied, and how to simultaneously maximize yield and water use efficiency through accurate estimation and monitoring of the various morpho-physiological indicators.

However, the large amount of data collected by spectroradiometer, which hold thousands of general wavebands between VIS and SWIR domains, often limits its high-throughput applications. To overcome these limitations, the specific and effective wavelengths in three parts of the spectrum can be used to calculate specific spectral reflectance indices (SRIs) using simple mathematical operations (normalized or ratio formulas). The advantage of using SRIs for assessment of plant parameters is that the plant parameters can be simultaneously assessed through different types of SRIs (e.g., simple ratio (SR), normalized spectral indices (NDSIs), perpendicular vegetation index (PVI), soil adjusted vegetation index (SAVI)). Most importantly, because several SRIs have been used to successfully estimate different parameters such as aboveground biomass and water content, leaf area index, gas exchange and transpiration rates, stomatal conductance, ion and pigment contents, carbon isotope discrimination, yield components, and grain yield in several field crops under either normal or abiotic stress conditions, efforts are going now to develop a new lightweight spectral sensor to detect these simple SRIs in the field. This new sensor will facilitate faster measurements in the field compared to the spectroradiometer used in the current studies and will bring down the current cost of equipment to a minimum level.

However, because the SRIs involved only 2–3 wavelengths and have been found to be affected by several factors such as environmental conditions, level of stress, growth stages, and crops types, there is still need for testing in different diverse environmental conditions in order to further validate known spectral reflectance indices (SRIs), to derive new and simpler SRI´s, and to develop monitoring models with wider applicability for indirectly assessment of stress-related plant parameters. Therefore, the one objective of this study was to evaluate whether SRIs (published and modified) could be used to indirectly estimate different soybean traits under different irrigation regimes and growth stages specifically, or at each growth stage across irrigation regimes or vice versa.

  • They assumed that crop growth and yield only affected by water content in soil and plant, neglecting other factors affect growth and crop yield (e.g., light, heat, soil nutrients).

Response: Many thanks for this remark. This study was conducted under field condition. Most importantly, under arid and semiarid conditions like Egypt and during summer season, the weather conditions (temperature, wind speed, and total and net solar radiations) remains fairly stable during the growing seasons of soybean as shown in Table 1.    

  • Finally, the Authors mixed up two types of sensing, remotes sensing, and proximal, neglecting the data preprocessing of both thermal and Vis-NIR spectroscopic measurements. The Authors used thermal and spectral data without indication of any preprocessing which is essential for spectral analysis. To my knowledge, raw spectral data is not recommended in spectral analysis especially with spectrophotometer such as tec5 that maybe have a spectral jump in the spectrum around 900 nm that required preprocessing to eliminate this jump. Furthermore, proximal sensing data required advanced data analytics- not simple linear regression- such as PLSR or other machine learning methods for better modeling.

Response: Many thanks for this comment. I totally agree with this important comment, especially regarding processing the data of canopy spectral reflectance. Because the full hyperspectral reflectance data often consists of hundreds of wavelengths that are redundant, noisy, and/or correlated by chance, it is necessary to apply the multivariate analysis such as PLSR or other machine learning methods for dealing with this type of data and overcomes the problems of overftting and multicollinearity that are inherent to hyperspectral datasets. These multivariate analyses are also very important to extract the most sensitive spectral band intervals and wavelengths associated with plant parameters from the full spectrum.

      However, in this study we focus on the ability of different constructed and published SRIs for assessing the growth, water status, and seed yield of soybean under different conditions. In generally, the SRIs often incorporate the most specific and effective wavelengths (one, two, or three specific wavelengths) and discard the majority of other wavelengths in the full VIS-NIR spectrum. These effective wavelengths are closely associated with the changes that take place in several biochemical and biophysical plant characteristics, such as plant pigment concentrations, photosynthetic efficiency, internal leaf structures, leaf area index, green biomass, vegetative vigour, and plant water status. For example, different leaf pigments are strongly related to the wavelengths in the VIS region, especially in the blue and red wavelengths. The wavelengths centered at 970, 1100, 1200, and 1240 nm provided insights for estimating different parameters related to plant water status.

Our future goal for this study is that we will apply the different multivariate analyses to identify the important wavebands associated with the crop variables of interest within the full VIS-NIR spectrum and compare the performance of the multivariate analysis based on full spectrum range with SRIs (which involves only 2-3 wavelengths) for assessing the growth, yield and water status of soybean under different irrigation regimes and at two growth stages. 

Specific comments

  • The title should be further improved and made clearer, more informative, and effective. Besides, it should reflect the objectives. The confusing terms should be avoided. The authors mixed between remote sensing and proximal sensing. So, I would suggest using the title ‘’Potential of Hyperspectral and Thermal Proximal Sensing for Estimating Growth Performance and Yield of Soybean Exposed to  Different Drip Irrigation Regimes Under Arid  Conditions’’.

Response: Many thanks for the suggestion of more informative and effective title for our manuscript. The title has been changed accordingly.  

  • The abstract does not explain well what was done, what was found, and what are the main conclusions.

Response: Many thanks for this remark. The abstract has been modified.

  • Keywords should be improved and made it easy for indexing and searching. There is nothing called proximal remote sensing; Proximal sensing and remote sensing are two different things. The contour map is not a well-known acronym in Proximal sensing and spectroscopic studies, use the appropriate acronym like 2-D-correlogram of R2.

Response: The keywords have been checked and modified.

  • The Introduction section should be improved and related to the more recent worldwide literature on the problem being investigated. The topic should be presented better and in a more general contest to allow its generalization to other areas of the world. Moreover, the Authors should briefly review better the most used methods to quantify the response of different vegetative growth traits and yield in different irrigation regimes base on optical proximal sensing and explain why they have chosen the proposed methods. Furthermore, the paper lacks a very clear and good justification for what is new and innovative about their approach.

Response: As we mentioned above, although several SRIs have great potential for assessing different plant parameters under normal and stress conditions, the performance of these indices for accurate estimation and monitoring of plant parameters has been found to be significantly affected by multiple factors related to the measurement conditions of canopy spectral reflectance such as level of stress, crop type, and crop phenological growth stages. Therefore, there is still need for testing the performance of SRIs for assessing plant parameters in different diverse environmental conditions in order to further validate known SRIs and/or derive new ones, which is the primary objective of this study. The further evaluation of SRIs under different conditions could help us to develop a new lightweight spectral sensor to detect these simple SRIs in the field. This new sensor will facilitate faster measurements in the field compared to the spectroradiometer used in the current studies and will bring down the current cost of equipment to a minimum level.

  • Line 23: Hyperspectral remote sensing, should be replaced with Proximal hyperspectral sensing. You did not use remote sensing in this research.

Response: Done

  • Lines 219-221: It is not clear how you did this. Did you use any spectral data preprocessing? Have you found any jump around 950 nm (linkage point of the two detectors) or noise at both edges of the spectrum in the tec5 spectrophotometer?

Response: In this study we focus on the performance of SRIs (which incorporate the most effective and sensitive wavelengths related to different biochemical and biophysical properties of the plant) for assessing plant parameters. The either published or new SRIs do not incorporate 950 nm as shown in Table 2. 

  • Lines 242-244: Please, be specific about the setup of the calibration (interval between scanning the white reference) and how you averaged the spectra?.
    Lines 316: Instead of contour map use ‘’ 2-D correlogram of the coefficients of determination…’’

Response: Many thanks for this remind. All information related to setup of calibration of spectral reflectance of canopy has been mentioned in details. In addition, the 2-D correlogram has been used instead contour map in all text. 

  • Lines 337: Explain the meaning of a, b, and c in the table at the footer?

Response: The a, b, and c in Table 3 indicate that the means followed by the same letter are not significantly different from one another based on Fisher’s least significant difference (LSD) test at p ≤ 0.05. This is already mentioned in the footer of the Table 3.

  • Lines 349-353: What about other factors that affect plant growth and crop yield? Please, rephrase this paragraph. 

Response: Under field conditions, we assume that all factors other than the factors under study (treatments of irrigation regimes in this study) have the same equal effect on plant growth. Fortunately, under arid and semiarid conditions like Egypt and during summer season, the weather conditions (temperature, wind speed, and total and net solar radiations) remains fairly stable during the growing seasons of soybean as shown in Table 1.     

  • Line 522: Use the range of wavelengths instead of bands, as you talked about specific wavelengths below, not bands.

Response: Done

  • Lines 728-732: This can be a future goal as this is not reflected in the measurement setup, the spectral data processing, as well as the advanced data analytics (e.g. machine learning and data fusion) which you did not apply in this research.

Response: Our future goal for this study is that we will apply the different multivariate analyses to identify the important wavebands associated with the crop variables of interest within the full VIS-NIR spectrum and compare the performance of the multivariate analysis based on full spectrum range with SRIs (which involves only 2-3 wavelengths) for assessing the growth, yield and water status of soybean under different irrigation regimes and at two growth stages. 

Reviewer 3 Report

The rationale for undertaking the research reported in the manuscript fits with the agrarian needs of poorly irrigated areas. The authors share with the reader insights from research on the possibility of remote evaluation of the quality of agricultural products in terms of their hydration and biomass efficiency. The authors place their observations in the context of the research conducted so far in the field of assessing the effectiveness of agricultural production in dry areas. The content of the manuscript is structurally closed. The only thing missing is tips on how to use the observations reported in this study operationally.
The schematic diagram in Figure 1 could be more carefully made - I did not understand what role element "Venture" plays.

Author Response

Reviewer#3

The rationale for undertaking the research reported in the manuscript fits with the agrarian needs of poorly irrigated areas. The authors share with the reader insights from research on the possibility of remote evaluation of the quality of agricultural products in terms of their hydration and biomass efficiency. The authors place their observations in the context of the research conducted so far in the field of assessing the effectiveness of agricultural production in dry areas. The content of the manuscript is structurally closed. The only thing missing is tips on how to use the observations reported in this study operationally.
The schematic diagram in Figure 1 could be more carefully made - I did not understand what role element "Venture" plays.

Response: We greatly appreciate your critical observations as well as your constructive and helpful comments. We hope that we could address your questions/comments by the explanations and revisions made in the manuscript. We believe that the manuscript is substantially improved after making the suggested revisions.

  • The schematic diagram in Figure 1 has been redrawn
  • The large amount of data collected by spectroradiometer, which hold thousands of general wavebands between VIS and SWIR domains, often limits its high-throughput applications. To overcome these limitations, the specific and effective wavelengths in three parts of the spectrum can be used to calculate specific spectral reflectance indices (SRIs) using simple mathematical operations (normalized or ratio formulas). The advantage of using SRIs for assessment of plant parameters is that the plant parameters can be simultaneously assessed through different types of SRIs (e.g., simple ratio (SR), normalized spectral indices (NDSIs), perpendicular vegetation index (PVI), soil adjusted vegetation index (SAVI)). Most importantly, because several SRIs have been used to successfully estimate different parameters such as aboveground biomass and water content, leaf area index, gas exchange and transpiration rates, stomatal conductance, ion and pigment contents, carbon isotope discrimination, yield components, and grain yield in several field crops under either normal or abiotic stress conditions, efforts are going now to develop a new lightweight spectral sensor to detect these simple SRIs in the field. This new sensor will facilitate faster measurements in the field compared to the spectroradiometer used in the current studies and will bring down the current cost of equipment to a minimum level.

Round 2

Reviewer 2 Report

The authors have addressed the detailed questions and comments I raised in my previous review. They have provided the detailed descriptions on the setup of the proximal sensing measurements, which now strengthen the experimental results.
From my point of view, the paper is almost suitable for publication in Sensors.